# In Vitro Validation of Antiparasitic Activity of PLA-Nanoparticles of Sodium Diethyldithiocarbamate against *Trypanosoma cruzi*

**DOI:** 10.3390/pharmaceutics14030497

**Published:** 2022-02-24

**Authors:** Johny Wysllas de Freitas Oliveira, Mariana Farias Alves da Silva, Igor Zumba Damasceno, Hugo Alexandre Oliveira Rocha, Arnóbio Antônio da Silva Júnior, Marcelo Sousa Silva

**Affiliations:** 1Immunoparasitology Laboratory, Department of Clinical and Toxicological Analysis, Centre of Health Sciences, Federal University of Rio Grande do Norte, Natal 59012-570, Brazil; johny3355@hotmail.com; 2Programa de Pós-Graduação em Bioquímica, Department of Biochemistry, Centro de Biociências, Universidade Federal do Rio Grande do Norte, Natal 59078-970, Brazil; hugo-alexandre@uol.com.br; 3Laboratory of Pharmaceutical Technology and Biotechnology, Department of Pharmacy, Federal University of Rio Grande do Norte—UFRN, Natal 59012-570, Brazil; marianafarias0011@gmail.com (M.F.A.d.S.); arnobiosilva@gmail.com (A.A.d.S.J.); 4Programa de Pós-graduação em Ciências Farmacêuticas, Department of Pharmacy, Centro de Ciências da Saúde, Universidade Federal do Rio Grande do Norte, Natal 59012-570, Brazil; 5Departamento de Engenharia de Materiais, Department of Engineer, Centro de Tecnologia, Universidade Federal do Rio Grande do Norte, Natal 59078-970, Brazil; igorzumba@ufrn.edu.br; 6Laboratório de Biotecnologia de Polímeros Naturais—BIOPOL, Departament of Biochemistry, Centro de Biociências, Universidade Federal do Rio Grande do Norte, Natal 59078-970, Brazil; 7Global Health and Tropical Medicine, Instituto de Higiene e Medicina Tropical, Universidade Nova de Lisboa, 1349-008 Lisbon, Portugal

**Keywords:** sodium diethyldithiocarbamate, *Trypanosoma cruzi*, Chagas disease, nanoparticle, nanoprecipitation

## Abstract

*Trypanosoma cruzi* is a protozoan parasite responsible for Chagas disease, which affects millions around the world and is not treatable in its chronic stage. Sodium diethyldithiocarbamate is a compound belonging to the carbamate class and, in a previous study, demonstrated high efficacy against *T. cruzi*, showing itself to be a promising compound for the treatment of Chagas disease. This study investigates the encapsulation of sodium diethyldithiocarbamate by poly-lactic acid in nanoparticles, a system of biodegradable nanoparticles that is capable of reducing the toxicity caused by free DETC against cells and maintaining the antiparasitic activity. The nanosystem PLA-DETC was fabricated using nanoprecipitation, and its physical characterization was measured via DLS, SEM, and AFM, demonstrating a small size around 168 nm and a zeta potential of around −19 mv. Furthermore, the toxicity was determined by MTT reduction against three cell lines (VERO, 3T3, and RAW), and when compared to free DETC, we observed a reduction in cell mortality, demonstrating the importance of DETC nanoencapsulation. In addition, the nanoparticles were stained with FITC and put in contact with cells for 24 h, followed by confirmation of whether the nanosystem was inside the cells. Lastly, the antiparasitic activity against different strains of *T. cruzi* in trypomastigote forms was determined by resazurin reduction and ROS production, which demonstrated high efficacy towards *T. cruzi* equal to that of free DETC.

## 1. Introduction

Chagas disease (CD) is a zoonosis caused by the flagellated parasite *Trypanosoma cruzi* that affects principally South America. CD has a spread of dispersion in many countries worldwide due to the immigration process in the risk zones carrying the disease, and this results in growing damage to non-endemic countries [1,2]. The pharmacological treatment for CD offered by the World Health Organization is benznidazole and nifurtimox. However, these drugs present high cytotoxicity, cause severe damage to the patients, and present low efficacy against chronic stages of CD due to their low bioavailability to reach the parasite inside the cells [3,4,5].

*T. cruzi*, the etiological agent of CD, has a complex life cycle and genomic diversity that allows it to survive and escape the immune system. Moreover, there is no efficient drug to treat the chronic stage of the disease. In previous studies, sodium diethyldithiocarbamate (DETC) demonstrated high antiparasitic activity against parasites *Trypanosoma cruzi* and *Leishmania* sp., which belong to the Trypanosomatidae family. The mechanism associated with its efficacy is based on the metal chelator activity and the stimulation of reactive oxygen species (ROS), causing damage to the parasite. The mechanisms are important to eliminate the parasite; however, they could cause some damage to the patient, as demonstrated in in vitro experiments [6,7,8,9].

One alternative to avoid the damage caused by DETC is using new technologies, and a promising alternative is the use of polymeric nanoparticles. Several studies already demonstrate the importance and benefits of nanosystems when applied to the treatment of parasitic diseases, reducing the cellular damage caused by some compounds, increasing the efficacy of drugs, and improving their bioavailability [10,11,12,13].

Poly-lactic acid (PLA), a biopolymer, is an aliphatic polyester approved by the Food and Drug Administration (FDA) for use as a biomaterial for therapeutic use. The PLA has important characteristics that have been demonstrated in the literature that make it useful as a polymer in polymeric nanoparticles. Several studies demonstrate its capacity to reduce cellular toxicity, increase permeability for compounds to enter in cells, and improve bioavailability [14,15,16,17,18,19]. These characteristics are important for the development of new treatments against *T. cruzi* in order to reduce the damage caused by new drugs and maintain their efficacy.

This work aims to associate the benefits of PLA nanoparticle systems demonstrated in the literature when coupled with the compound sodium diethyldithiocarbamate, which demonstrated, on a free stage, high efficacy against *T. cruzi.* Moreover, this association aims to reduce any damage that DETC can cause to cells, improve its bioavailability, and maintain the compound efficacy.

## 2. Materials and Methods

### 2.1. Cells and Parasites Maintenance

The cellular lineages RAW (ATCC number TIB-71), derived from macrophages, 3T3 (ATCC CRL-1658), derived from fibroblasts, and VERO (ATCC CCL-81), derived from epithelial kidney cells, were generously donated by the Biopolymers Laboratory (Natal, RN, Brazil) and were grown in DMEM medium (Cultilab, Campinas, SP, Brazil) supplemented with fetal bovine serum (FBS) at 10% (*v*/*v*) (Cultilab, Campinas, SP, Brazil) and 100 U/mL penicillin and 100 μg/mL streptomycin. The cells were incubated at 37 °C with 5% CO_2_, and the medium was changed every three days for maintenance. The cells were further subculture at 80% confluence using a cell scraper (RAW cells) or trypsin/EDTA (3T3 and VERO cells).

Different strains of *T. cruzi* TcI (strain Dm28c and Bolivia) and TcII (strain Y) were cultured in an LIT (Liver Infusion Tryptose) medium supplemented with fetal bovine serum (FBS) at 10% (*v*/*v*) and 100 UI/mL of streptomycin/penicillin. All strains were generously donated by the Laboratório de Parasitologia (FCFAr)*,* Araraquara, Brazil. The culture of epimastigote forms of *T. cruzi* was maintained at 27 °C in a BOD (incubator chamber, ASP, SP-500).

To obtain trypomastigote forms of different strains of *T. cruzi*, we utilized the method described by Contreras et al. [20]. Briefly, the parasites in medium LIT, after 5 to 7 days, achieved the stationary phase; then, the parasites were washed with PBS 1× pH 7.4 three times and resuspended in TAU medium. After 2 h, we added 10 µM of proline and sodium bicarbonate 1.4 M in the proportion of 1:4 (*v*/*v*). Then, the culture was kept for four days without movement at 27 °C in a BOD. The metacyclogenesis was confirmed by optical microscopy (Biolab Brasil, São Paulo, Brazil).

### 2.2. Preparation of PLA Nanoparticles of DETC Using Nanoprecipitation

The poly-lactic acid (PLA) nanoparticles were prepared using nanoprecipitation with solvent evaporation methodology. The organic phase was composed of acetone PA (dielectric constant ε 20.6) and ethanol PA (ε 24.6) in a volumetric ratio of 80:20. We used PLA at 0.5% (*w*/*v*) as a polymeric matrix, and sodium diethyldithiocarbamate (DETC) as the drug of choice to be incorporated into the system. The aqueous phase used polyvinyl alcohol (PVA) as a surfactant with a molar mass of 61.000 g/mol and a viscosity of 9–11 mPa·s at a concentration of 0.25% (*w*/*v*). A volume of 6 mL of the organic phase was injected into 14 mL of the aqueous phase, maintaining the volumetric proportion ratio of 30:70 in a continuous flow of 1 mL/29 s. The phases were previously filtered through a nylon membrane of size 0.22 µm and a cellulose acetate membrane of size 0.45 µm, respectively. Obtaining the nanoparticles was possible by completely removing the organic solvents under magnetic stirring at 720 rpm at 25 °C, overnight.

### 2.3. Physical and Chemical Characterization of DETC Nanoparticles

The nanoparticles containing DETC were prepared using the solubilization in the organic phase component of ethyl alcohol (EtOH). DETC was added in 1:12 drug/polymer proportions, which corresponds to 8.2% (*w*/*w*) of the drug in the system. After adding DETC, the experiment followed the same procedure for obtaining nanoparticles without the drug, mentioned in the former section.

#### 2.3.1. Determination of Particle Diameter, Zeta Potential, and pH

In order to characterize the systems obtained, the particle diameter was determined using the light scattering methodology (Dynamic Light Scattering) in a ZetaSizer NanoZS analyser model (82 Brookhaven, Malvern, UK), at angle of detection of 173°. The measurement of the zeta potential was performed with the same equipment through electrophoretic mobility. Undiluted samples were analyzed using the pH meter (Gehaka, PG 1800, São Paulo, Brazil. All analyses were performed in triplicate at room temperature 25 ± and data are expressed as mean ± standard deviation (SD).

#### 2.3.2. Encapsulation Efficiency and Drug Loading

To determine the encapsulation efficacy and drug loading, aliquots of 1 mL of the nanoparticle suspension samples with and without DETC were prepared. Then, the samples were centrifuged in an ultrafiltration device (Sartorius^®^, Vivaspin 2, Ultra-15MWCO 10 kDa) (Eppendorf, Hamburg, Germany) using an Eppendorf^©^ 5804R refrigerated centrifuge (Sartorius AG, Argentina, Germany) 16.9 RCF (g) for 60 min at 4 °C. After obtaining the filtrate, the NP were analyzed via UV–Vis spectrophotometry (Thermo Fisher, Whalthan, MA, USA) using a spectrophotometer and a UV–Thermo Fisher Scientific evolution 60S spectrophotometer (Thermo Fisher, Whalthan, MA, USA). With the aid of the linear regression equation extracted from the standard curve built under the same conditions of analysis, the content of the drug incorporated in the systems was determined. Encapsulation efficiency (EE) and drug loading (DL) were calculated using Equations (1) and (2).
(1)$$EE\left(\%\right)\;=\frac{\left(totaldrug-drugsupernatant\right)}{totaldrug}\times 100\;$$
(2)$$DL\left(\%\right)\;=\frac{\left(totaldrug-nanoparticles\right)}{totalnanoparticles}\times 100\;$$

#### 2.3.3. Infrared Absorption Spectroscopy (FTIR-ATR)

Spectroscopic analyses in the infrared region were carried out to investigate the drug interaction level with the structural components of the systems or the possible components’ chemical changes. For this, the absorption spectra in the infrared region of the isolated structural components, the different formulations, and the respective physical mixtures of the same composition were obtained. The equipment used was a Transformed Fourier Infrared (FTIR)—ATR, SHIMADZU IR Prestige 21 (Shimadzu, Tokyo, Japan). The spectra were recorded with 20 scans at a resolution of 4 cm^−1^ wavenumbers between 4000 and 500 cm^−1^.

#### 2.3.4. Atomic Force Microscopy (AFM)

The nanoparticles shape and size were observed using AFM images. The dispersions were previously dried in a desiccator for 24 h. The measurements were performed using AFM (SPM—9700, Shimadzu, Kloten, Switzerland) at 25 °C in a non-contact cantilever, with digitalization at 1 Hz.

#### 2.3.5. Scanning Electron Microscopy (SEM)

Samples analyzed using field emission scanning electron microscopy (SEM) (Model augira, Brand Carl Zeiss, Oberkochen, WB, GER) were fixed on a conductive adhesive surface and covered with approximately 20 nm gold in a BAL-TEC sputter coater (Bal-Tec, Scotia, USA), model SCD 005, in order to make them conductive. Analyses were performed with 25 kV of electron acceleration voltage, using images formed by the secondary electron detector (ETD) (Model augira, Brand Carl Zeiss, Oberkochen, WB, GER).

### 2.4. In Vitro DETC Release Study

The in vitro release assay was performed on Franz diffusion cells. The dissolution medium employed was a 50 mM monobasic potassium phosphate-buffered solution (pH = 7.4). The systems remained in constant agitation at 360 rpm and 37 ± 2 °C during the whole experiment. At predetermined time intervals (30, 60, 90, 120, 180, 300, and 360 min), samples of aliquots of the release medium were collected through the sidearm of the Franz cell. The same volume of freshly buffered solution replaced the medium to maintain the sink conditions. Through the linear regression equation of the standard drug curve, the drug released from the nanoparticles was determined using UV–Vis spectrophotometry. The drug solution was prepared at concentrations equivalent to those present in the nanoparticles (200 μg/mL).

### 2.5. Toxicity against Cellular Lineages

To evaluate if the nanoparticles could cause toxicity, we performed MTT assay against three cellular lineages: RAW (ATCC number TIB-71), derived from macrophages, 3T3 (ATCC CRL-1658), derived from fibroblasts, and VERO (ATCC CCL-81), derived from epithelial kidney cells. The assay was performed according to the method described by Mosmman [21]. Briefly, the cells were plated in a 96-well plate in concentration of 5 × 10^3^ cells/well and grown for 24 h in culture conditions. Hereafter, different concentrations of nanoparticle systems containing DETC (22 to 132 µM) and nanoparticle systems without DETC (22 to 132 µM) were applied in a 96-well plate and kept in culture conditions for 24 h. After 24 h, the culture medium was withdrawn, and to it was added 100 μL of MTT at a concentration of (1 mg/mL dissolved in DMEM medium without FSB). Then, the cells were incubated for 4 h at 37 °C and 5% CO_2_. Posteriorly, the culture supernatant was discarded, and to it was added 100 μL/well of ethanol to solubilize the formazan crystals. The absorbance was measured with the equipment Epoch microplate spectrophotometer (Biotek Instruments Inc., Winooski, VT, USA) at λ = 570 nm. Cell viability was calculated in relation to the negative control using the formula: % viability = (Atest/A Control) × 100, in which Atest corresponds to the absorbance of the experimental group, and A Control corresponds to the absorbance of the negative control.

### 2.6. Fluorescence Nanoparticle Production and Capacity to Penetrate Cells

To evaluate if the nanoparticles carrying DETC can penetrate cells, we produced fluorescent nanoparticles to which the cells were exposed. Briefly, the nanoparticles were lyophilized; 30 mg of lyophilized nanoparticles were added to 1 mg of fluorescein and dissolved in 10 mL of PBS 1× pH 7.4 under agitation for 1 h. After this period, the excess of fluorescein was removed by dialysis using membranes of 6 kDA for 4 h under agitation, and during this period the water was changed 3 times to facilitate dialysis. After dialysis, the nanoparticles were lyophilized again for the next steps.

To analyze if stained nanoparticles can penetrate the cells, we utilized the cellular lineages RAW and VERO. The cells were plated in glasses inside 24-well plates at a concentration of 3 × 10^4^ cells/plate and kept for 24 h in DMEM medium supplemented with SFB at 37 °C and 5% of CO_2_. Hereafter, the stained nanoparticles were added for different periods (15 min, 30 min, 1 h, 2 h, 12 h, and 24 h) to evaluate if the time was an important factor when analyzing if the particles can penetrate the cell membrane. After this, the glasses containing the cells were fixed with methanol for 20 min, removed, and washed 3 times in cold PBS 1× pH 7.4. The images were obtained from different fields, randomly selected, which were analyzed using the NIS Elements AR v. 4.00.03 64-bit software (Nikon (2011), Melville, NY, USA) in blue spectra.

### 2.7. In Vitro Antiparasitic Activity against Different Strains of Trypanosoma cruzi

Trypomastigote forms of different strains of *T. cruzi* were used to perform this assay. Here, the parasites were plated in 96-well plates at a concentration of 1 × 10^7^ parasites/mL and exposed to different concentrations of DETC nanoparticles (11 µM at 132 µM) for 24 h at 28 °C. After exposure, the parasites were submitted to a colorimetric viability assay using resazurin, as previously used by the author of [9]. The results were plotted as a percentage (%) of parasite deaths. The selective index was calculated based on results of the IC_50_ of cell lines exposed to DETC and the IC_50_ of DETC against trypomastigote forms of *T. cruzi*. Benznidazole was used as a positive control in the antiparasitic test and the IC_50_ was measured too.

### 2.8. Induction of ROS Production by Parasites Exposed to DETC Nanoparticles

All strains of *T. cruzi* had their reactive oxygen species (ROS) production after treatment with nanoparticles of DETC detected using the marker 2’,7´-dichlorofluorescin diacetate (Sigma, Saint Louis, MO, USA). Briefly, the different strains of *T. cruzi* on epimastigote forms were treated with DETC nanoparticles at a concentration of 44.00 µM for 24 h in the same conditions described previously. After treatment, the parasites were centrifuged at 2000 rpm for 6 min at 4 °C and washed with PBS pH 7.4 twice. Afterward, these parasites were loaded with 10µM of 2′,7′-dichlorofluorescin diacetate and maintained in a dark room for 45 min. The endogenous ROS hydrogen peroxide (H_2_O_2_) was used as a positive control of 0.5 mM. The ROS production was determined by the increase in fluorescence caused by the conversion of the probe and read with the equipment GloMax^®^ Discover Microplate Read model GM 3000 (GloMax, Fitchburg, WI, USA) at λex = 490–530 nm.

### 2.9. Statistical Analysis

All experiments were executed in triplicate and as independent systems. The data were processed, and the results are presented in the form of mean ± standard deviation. The data were submitted to the normality test of Shapiro–Wilk. The parametric results for 3 or more groups using the ANOVA test were further processed with Tukey’s post hoc test, and for 2 groups with Student’s *t* test. Tests were performed using the software GraphPad Prism v. 7.0 (2016, GraphPad, San Diego, CA, USA) and P.A.S.T v. 2.17 (2012, PAST, Oslo, Norway).

## 3. Results

### 3.1. Physical and Chemical Properties of Nanoparticles of DETC via Nanoprecipitation

DETC was incorporated at a 1:12 ratio of drug/polymer using a concentration of 0.5% PLA and 0.25% PVA to obtain NPD nanoparticles via the nanoprecipitation method (Figure 1A). The physicochemical properties of different drug-free (NPB) and drug-loaded (NPD) nanoparticles are shown in Table 1. The formation of a single particle family could be observed in the nanoparticle diameter distribution profile according to the light scattering intensity (Figure 1B). The quantitative analysis revealed that NPD resulted in an entrapment efficiency of around 72% and a drug loading level of 3.63% that corresponds to a final drug concentration of 200 µg/mL. The structure observed in the results presented in Figure 1C,D represents the structure of NPB and NPD. It is possible to observe a uniform size of the nanoparticle structure and a low dispersion with regard to their conformation in both the AFM and SEM images.

### 3.2. Infrared Absorption Spectroscopy (FTIR-ATR)

The FTIR spectra recorded for pure compost PLA and DETC, the physical mixture (PM) of PLA+DETC, and the nanosystems NPB and NPD are shown in Figure 2. In the PLA spectrum, it is possible to observe the presence of the stretching of the C=O bond of the carbonyl groups at 1747 cm^−1^ and the –CH and –CH_3_ bands at 2995 cm^−1^ and 2945 cm^−1^ [22]. In the DETC spectra, between 2850 and 3000 cm^−1^, C–H stretches are evident. The peak at 1415 cm-1 demonstrated the absorption for group N–CS2. In 1128 and 2063 cm^−1^, we can observe the stretching C=S and, between 600 and 1100 cm^−1^, we can observe the stretching C–S, C–C, C–N, CH_2_, and CH_3_. A region below 600 cm^−1^ corresponds to altered torsions [23,24,25]. The physical mixture presented important differences compared with NPD, mainly the intensity of the bands and the bathochromic shift of the C=O bond of the carbonyl of ester band and the overlapping by DETC bands that occurred in the physical mixture. Furthermore, in PM, which resulted from a mixture of 1:1 (*w*/*w*) of PLA and DETC, it is possible to observe a reduction in the intensity of the DETC peaks when compared with free DETC and PLA peaks, which are not clearly visualized; however, the reduction in the DETC peaks indicates the presence of another compound.

### 3.3. In Vitro Drug Release

The in vitro drug release assay is fundamental for understanding the physicochemical properties and mechanisms that influence drug release. Figure 3A shows the DETC release behavior from the DETC solution (DS) and from NPD. At 360 min, DS showed a drug release of 48.3%, while NPD showed a release of 15.6% at the same time. It was observed that the release of DETC from the nanoparticles was the slowest, representing a more controlled release than the drug solution. To determine the DETC release mechanism in NPD, mathematical kinetic models of the zero-order, first-order, Bhaskar, Freundlich, and parabolic types were applied [26,27,28]. After an analysis of the release determination coefficients, the parabolic diffusion model ((M_t_/M_∞)_/t = kt^−0.5^) showed the best fit (R^2^ > 0.99) (Figure 3B).

### 3.4. Cellular Toxicity of DETC Nanoparticles

Three cell lines were used in the MTT assay to analyze the cytotoxicity of DETC nanoparticles. In Figure 4, we observed that RAW, 3T3, and VERO present a consistent curve when tested against different concentrations of nanoparticles. In Figure 4A, just the high concentration of 132 µM of DETC nanoparticles caused a significant reduction in the cellular viability of VERO cells; on the other cell lines, we observed a constant result in cellular viability. Furthermore, we performed the test against the nanoparticles without DETC (Figure 4B), and possibly observed a similar profile to the DETC nanoparticles, except in the high concentration of 132 µM against the VERO cells, where the viability was reduced to 60%. During all experiments, DETC nanoparticles and nanoparticles without DETC did not cause high lethality to cell lines.

### 3.5. Capacity of Permeability Membrane Carrying Drugs

To evaluate if nanoparticles with DETC could enter easily inside cells, we performed an assay to demonstrate this capacity. The results presented in Figure 5 demonstrate when nanoparticles were marked with FITC and put in contact for 24 h with two different cell lines (RAW and VERO). In the image is shown the presence of fluorescence inside cells after the process, and the cell line RAW demonstrates more intensity in fluorescence when compared with cell line VERO.

### 3.6. Antiparasitic Activity against Trypanosoma cruzi

The NPD antiparasitic activity was tested against different strains of *T. cruzi* in trypomastigote forms (Figure 6). The results show the inhibition curve of the parasite with an increase in the concentration of nanoparticles and how the different strains respond differently to the compound. The strains Dm28c and Y were demonstrated to be more susceptible to DETC nanoparticles when compared to the results with the strain Bolivia. In addition, we determined the IC_50_ based on the results for antiparasitic activity that can be observed in Table 2. In these results, beyond demonstrating the efficacy of the PLA-DETC nanoparticles, we demonstrate a simple comparison with the results of a commercial drug used against *T. cruzi*, benznidazole, whose IC_50_ against these different strains is shown. It was possible to observe that DETC nanoparticles presented a lower IC_50_ than benznidazole.

### 3.7. Stimulation of ROS Production by Trypanosoma cruzi after Exposure to DETC Nanoparticles

To understand the mechanisms associated with DETC that cause the death of the parasite, the ROS production inside the parasite was analyzed and the results are shown in Figure 7. The nanoparticles of DETC interfere in ROS production by the parasites. In addition, the strains suffer differently against exposure to DETC. Strains Y and Dm28c were less affected by the nanoparticles than the strain Bolivia, which presented the highest ROS production. Furthermore, the parasites, when exposed to nanoparticles without DETC, demonstrated the capacity to increase ROS production.

## 4. Discussion

In this study, using biodegradable PLA nanoparticles as a promising nanotechnological approach to improve DETC activity was investigated. All formulations had small and monodispersible particle diameters. The average size being smaller than 200 nm (PdI < 0.3) demonstrates that the formulation led to the development of a narrow, uniform and single-mode particle size that is considered to be suitable to overcome biological barriers [29]. The distribution of nanoparticle diameters presents the following profile (Figure 1B), which confirms the formation of a single particle family (characterized by a unimodal particle distribution) located in the range between 100 and 200 nm. The increase in particle size and the change in pH is indicative of DETC incorporation into NP, which is confirmed by the EE and DL measurements [30].

Luize Mazur (2018) revealed similar physicochemical results, showing a particle size of around 190 nm (PdI < 0.2) and negative zeta potential. However, the DETC encapsulation efficiency was close to 90% in solid lipid nanoparticles systems. Submicrometric sizes with a narrow distribution and satisfactory encapsulation efficiency are important in the development of nanosystems, especially for their biological application [31]. Furthermore, Assolini et al., in 2020, using double emulsion to develop a DETC-Beeswax-CO Nps using copaiba oil, obtained a nanoparticle with a size of 200 nm. The system developed in this study presents a size of around 168 nm, and the technique applied to encapsulation and the biopolymer used can be adapted to change the size of nanoparticles. In addition, the reduction in size could be an important point of treatment and lead to the elimination of parasites due to the capacity to enter cells and pierce their barriers [32].

In Table 3 below, we highlight some nanoparticles developed using DETC, a compound with different nanosystems, applied against the *Tryponosomatidae* family, specifically against *Leishmania* sp. However, in the parasitology field associated with nanoparticles, DETC was presented in a few studies for its application.

In order to investigate the interaction between the drug and the structural components, a spectroscopic analysis in the infrared region was performed. Comparing the spectra of physical mixtures in the 1:1 (*w*/*w*) ratio of PLA/DETC with NPD showed a relevant difference. The physical mixture (PM) was prepared by using a PLA/DETC ratio of 1:1 *w*/*w*, which exhibited an FTIR spectrum similar to that recorded for the pure DETC. However, it is possible to observe important differences, such as the intensity of DETC bands and, significantly, the bathochromic shift of the ester band from PLA and the overlapping of the DECT bands. This phenomenon did not occur with drug-loaded nanoparticles, which corroborates the identification of a different drug–polymer interaction. Through comparisons of NPB vs. NPD in the FTIR spectrum, it is possible to identify a high intensity of the ester band at 1750 cm^−1^ in the drug-loaded nanoparticles [33].

The similarity of PM and free DETC can be associated with the hydrophobicity of DETC. When dissolved with PLA, this characteristic can be expressively elucidated by the results that we have obtained. In addition, we have clearly observed the reduction in the intensity of DETC peaks in the region of 2800–3000 cm^−1^ and around 800–1000 cm^−1^ when comparing PM and free DETC. These results are observed in other works using DETC, and the mixture is characterized by a reduction in the peaks of the compound [31,32].

Cordeiro et al. (2021) showed the FTIR spectrum of the free DETC, and the free DETC spectra were similar to this study, with characteristic peaks at 837, 909, and 985 cm^−1^, representing the C-S bond, 1070 and 2104 for C=S, and also 1423 cm^−1^ for N–CS_2._ Furthermore, DETC was co-encapsulated with 4-nitrohalcone in beeswax nanoparticles, and a greater intensity in the peaks of these compounds was observed according to the amount present in the formulation [25]_._ Furthermore, the same peaks in the FTIR were observed by Mazur et al. when developing nanoemulsions of DETC and beeswax, but they were less intense than our findings because they used a shorter concentration of DETC than in this experiment [31].

The experimental data show that DS and NPD exhibit different release behavior. In NPD, the DETC release rate was slower and more controlled compared to the pure drug solution. We suggest that the association with the polymeric shell causes a delay in the molecule dissemination. According to this, and associated to the adjustment in the parabolic kinetic model, it is implied that the release rate was controlled by diffusion dependent on the drug loading level [34].

DETC nanoparticles demonstrated low toxicity against different cell lines in Figure 4, and the results were similar to NPB. These results are associated with the controlled release of the drug and demonstrate the importance of nanoparticles for reducing toxicity, as observed in other studies, whereby the controlled release of drugs reduced the damage induction because of the minimal presence of the compound in cells. As observed in the results in Figure 3, the release of DETC was minimal when encapsulated, and that was essential to this result [35,36,37,38,39]. In addition, we observed that other studies using different nanosystems associated with DETC present a reduction in cellular toxicity compared with free DETC, and the results achieved by other researchers are similar at ours, with just a little toxicity when using elevated concentrations of nanoparticles with DETC [31,32].

In addition, associated with the test of cytotoxicity, we performed an assay using stained NPD with FITC (Figure 5). We observed that nanoparticles can enter the cells, and this is an important point when validating tests against Chagas disease in the chronic stage, principally when the objective is the production of new drugs using new technologies is to improve the efficacy and safety of organisms [40,41].

The nanosystem proprieties that we produced also demonstrate its capacity to enter into cells, since the PLA coating presents characteristics that facilitate association with membranes [42,43]. These characteristics are possible due to the polarization of PLA nanoparticles, or polymeric nanoparticles, aiding their attachment to membranes and helping to pierce through them. Furthermore, PLA is a component approved by the FDA, and its presence in the literature as a material against *T. cruzi* is positive, with good prospects regarding its ability to improve efficacy and safety [14,44]. Furthermore, some studies have demonstrated the capacity of PLA nanoparticles to penetrate the cellular membrane and barriers [45,46,47].

Moreover, DETC nanoparticles present a high antiparasitic activity, as shown in Figure 6. PLA-DETC nanoparticles were tested against different strains of *T. cruzi* (Dm28c, Y, and Bolivia), and these strains present variable genetic patterns, evidencing that DETC nanoparticles can affect different variants of the genome of the parasite. The genetic variation resulted in different profiles of parasite death during the experiment after exposure to the compound, and this variation is normal because the alteration in genetic patterns can result in alterations in resistance [48,49].

Furthermore, when comparing the results achieved here with free DETC, we observed a similar profile of IC_50_ for free DETC and PLA-DETC nanoparticles against *T. cruzi.* These results indicate that the encapsulation of DETC did not reduce the capacity to eliminate the parasite, although there was a controlled release of drug by the nanosystem [9].

We found that the parasite, when exposed to PLA-DETC, increased ROS production (as visualized in Figure 7). This is due to the DETC present in an amine group, which acts in ROS production, and this intensifies the ROS production inside the parasite. The accumulation of reactive species induces damage to the parasite and its posterior death [7,8,9]. The literature already demonstrates the capacity of DETC to increase ROS production in its free form when the parasite is exposed to it. Furthermore, the literature demonstrated the importance of ROS production to the death of the parasite via the accumulation of damage caused by molecular oxidation [6,7,8,9,32].

Recently, a study using disulfiram, in which DETC is produced metabolically during degradation, demonstrated its promise in the treatment of *T. cruzi*, and demonstrated high activity, showing that it was capable, when associated with BNZ, to lead a better result against the parasite [50]. These findings demonstrate the importance of using the metabolic compound of disulfiram in new tests and mechanisms of delivery, which would also increase the probability of the compound DETC, when associated with nanoparticles, being used as a new drug for the treatment of Chagas disease.

This study demonstrates a possible new drug that could be used in CD treatment, presenting results that demonstrate the great potential of DETC nanoparticles, which are associated with viability for cells and high levels of damage to parasites. However, in vivo studies in preclinical models will be necessary to enhance the information presented in this work, such as toxicological profile analysis, antiparasitic activity analysis, and pharmacokinetic profiling, which are the important next steps of this study.

Furthermore, CD is a disease that principally affects poor regions of the world, making necessary the use of systems, models, and technologies with low cost to guarantee that a newly discovered drug can help the regions most affected by the disease. In this study, the product developed, PLA-DETC nanoparticles, can alleviate these concerns. The technology and the methods in this study involve low costs when fabricating the PLA-DETC nanosystem. In addition, both main compounds are cheap to acquire, and the production system used was easy and fast.

## 5. Conclusions

The PLA-DETC nanosystem presents high stability and a small size of around ~164 nm, which was verified by SEM, AFM, and DLS analyzes. The PLA-DETC nanoparticles demonstrate controlled drug release, which was proven in an in vitro study. Moreover, that capacity is associated with the low toxicity of the system against three cell lines RAW, 3T3, and VERO, which was verified in vitro, demonstrating that the nanoparticles are safe and able to reduce DETC toxicity. In addition, the capacity of nanoparticles to enter the cells was proven by fluorescence, using nanoparticles stained with FITC. This is an important mechanism for the treatment of the chronic stage of CD due to the increase in the bioavailability of the compound. In addition, PLA-DETC presented high antiparasitic activity against different strains (Y, Dm28c, and Bolivia) of *T. cruzi* in trypomastigote forms, and also caused an increase in the ROS production of the parasites, showing that exposure to the system can cause damage and posterior death. The results demonstrated the importance of the PLA-DETC nanosystem as a possible alternative to the treatment of Chagas disease, especially for its use in the chronic stage of the disease, which does not have an efficient treatment.

## Figures and Tables

**Figure 1 pharmaceutics-14-00497-f001:**
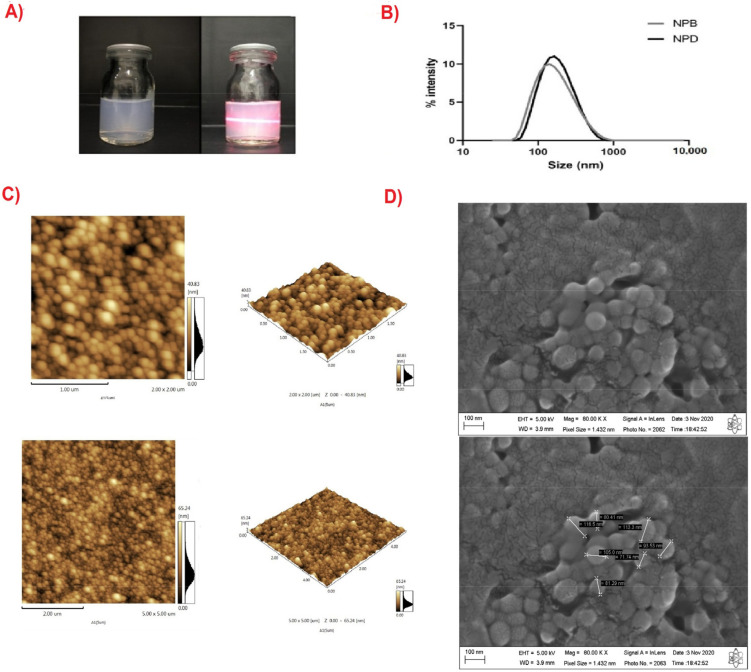
DETC-loaded nanoparticles. (**A**) The red-light scattering highlights the “Tyndall effect”. (**B**) Particle size intensity distribution graph of DETC-loaded nanoparticles and drug-free nanoparticles. (**C**) Atomic force microscopy analysis of drug-free and DETC-loaded nanoparticles. (**D**) Scanning electron microscopy analysis of drug-free and DETC-loaded nanoparticles.

**Figure 2 pharmaceutics-14-00497-f002:**
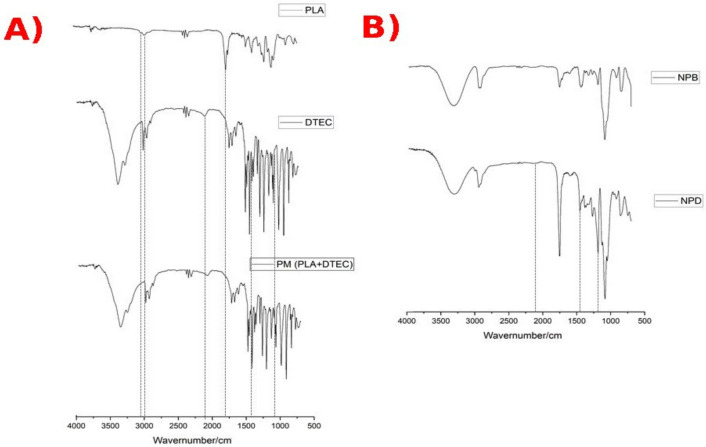
FTIR-ATR analysis. (**A**) Spectrum in region IV of the individual pure compounds and the compound physical mixture. (**B**) Spectrum in region IV of the drug-free (NPB) and DETC-loaded nanoparticles (NPD).

**Figure 3 pharmaceutics-14-00497-f003:**
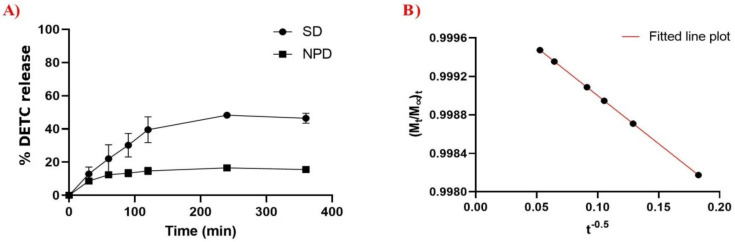
Experimental in vitro release profile. (**A**) Analysis of DETC solution (DS) and DETC-loaded nanoparticles (NPD) at the same time intervals and conditions. Results presented as mean ± standard deviation in an independent triplicate system. (**B**) Parabolic model of NPD during performed assays. The dots are the points obtained by equation to determinate the R score.

**Figure 4 pharmaceutics-14-00497-f004:**
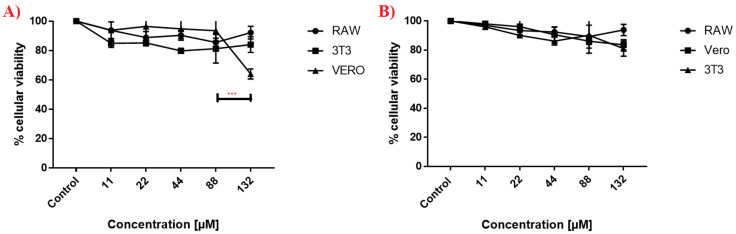
Viability of different cell lines 24 h after being exposed to different nanosystems. The cell lines RAW, 3T3, and VERO were exposed to different concentrations of DETC-loaded nanoparticles and drug-free nanoparticles for 24 h. (**A**) DETC-loaded nanoparticles (NPD); (**B**) drug-free nanoparticles (NPB). Results presented as mean ± standard deviation in an independent triplicate system and for the statistical analysis of the ANOVA test, together with Tukey’s post hoc test; *p* < 0.001 (***). In order to verify differences, the profile of each strain was compared against others treated with the same concentration of DETC nanoparticles and the controls.

**Figure 5 pharmaceutics-14-00497-f005:**
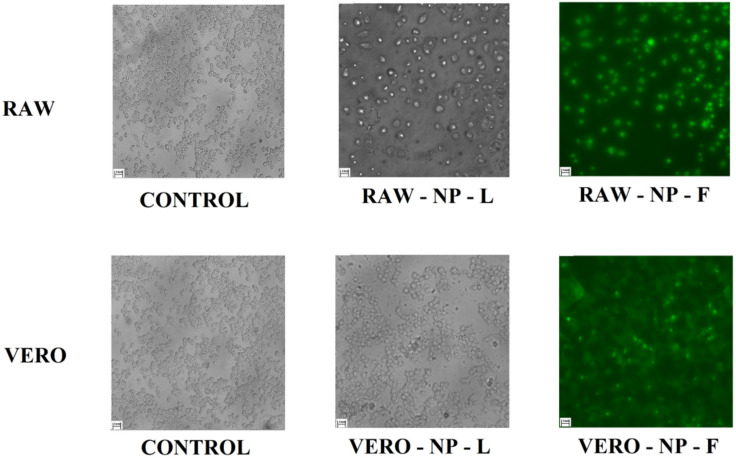
Fluorescent microscopy of different cell lines 24 h after being exposed to stained DETC-loaded nanoparticles. The cell lines RAW and VERO were exposed to DETC nanoparticles stained with FITC to evaluate if nanoparticles can enter inside cells, the scale bar is 0.1 mm/cm.

**Figure 6 pharmaceutics-14-00497-f006:**
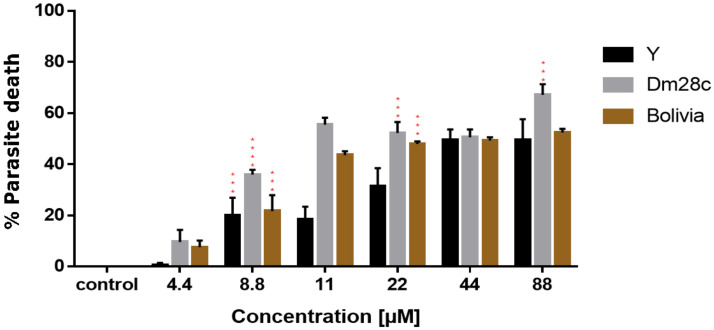
Trypanocidal activity of DETC nanoparticles against different strains of *Trypanosoma cruzi*. The parasites were exposed to different concentrations of DETC-loaded nanoparticles (4.4 µM to 88 µM) for 24 h. Results are presented as mean ± standard deviation of the percentage of parasitic inhibition in an independent triplicate system and for the statistical analysis of the ANOVA test, together with Tukey’s post hoc test; *p* < 0.001 (***), *p* < 0.0001 (****). In order to verify differences, the profile of each strain was compared against others treated with the same concentration of DETC nanoparticles and the control.

**Figure 7 pharmaceutics-14-00497-f007:**
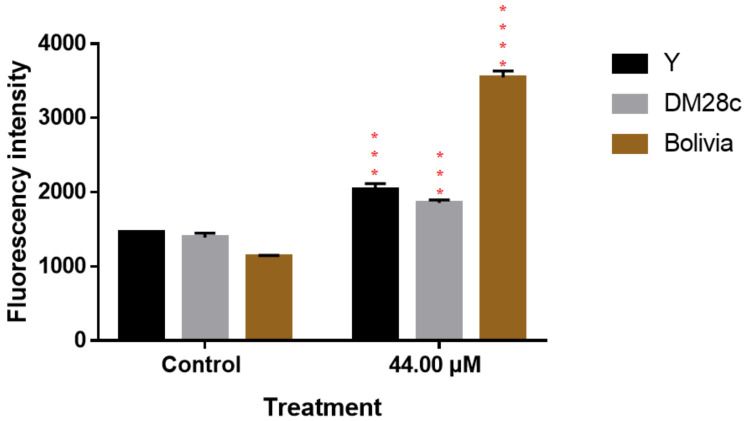
ROS production by *Trypanosoma cruzi* 24 h after being exposed to DETC nanoparticles. The parasites were exposed to a concentration of 44.0 µM of NPD for 24 h. Results are presented as mean ± standard deviation of the percentage of parasitic inhibition in a triplicate system and for the statistical analysis of the ANOVA test, together with Tukey’s post hoc test (*p* < 0.001 (***); *p* < 0.0001 (****). In order to verify differences, the profile of each strain was compared against the control.

**Table 1 pharmaceutics-14-00497-t001:** Physically linked properties of the blank (NPB) and DETC-loaded nanoparticles (NPD). Results presented as mean ± standard deviation in an independent triplicate system.

Sample	Size (nm) ± SD	PdI (nm) ± SP	ZP (mV) ± SP	pH	EE (%)	DL (%)
NPB	143.8 ± 2.73	0.154 ± 0.08	−21.80 ± 2.75	4.10	-	-
NPD	164.7 ± 2.96	0.221 ± 0.02	−19.50 ± 5.15	7.9	72.65	3.63

Note: SD (standard deviation); PdI (polydispersity index); ZP (zeta potential); EE (encapsulation efficiency); DL (drug loading).

**Table 2 pharmaceutics-14-00497-t002:** IC_50_ of DETC nanoparticles against different strains of *T. cruzi* after 24 h of exposure. Results are presented as mean ± standard deviation in an independent triplicate system and for the statistical analysis of the ANOVA test, together with Tukey’s post hoc test, in which different letters indicate statistically significant differences at *p* < 0.05.

Strain	IC_50_ of Compounds against *T. cruzi*	
	NPD (µM)	Benz * (µM)
Dm28c	15.47 ± 2.71 ^a^	70.58 ± 6.87 ^c^
Y	45.15 ± 5.44 ^b^	85.24 ± 5.22 ^d^
Bolivia	47.89 ± 3.98 ^b^	79.78 ± 6.18 ^c^

***** Note: Benz (benznidazole). a, b, c and d indicate statistical difference among groups.

**Table 3 pharmaceutics-14-00497-t003:** Comparative table of findings in the literature on the use of nanoparticles with DETC against the *Trypanosomatidae* family.

Name	Nanosystem	Size Obtained (nm)	Zeta (mV)	Efficacy Encapsulation (%)	Ref.
DETC-Beeswax-copaiba	Double Emulsion	~190	~−42	~90	[31]
DETC-Beeswax-CO	Double Emulsion	~200	~−44	~87	[32]
PLA-DETC	Precipitation	~164	~−20	~78	-

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
