# Peer review of "In Vitro Validation of Antiparasitic Activity of PLA-Nanoparticles of Sodium Diethyldithiocarbamate against Trypanosoma cruzi"

_pharmaceutics, 2022, doi:10.3390/pharmaceutics14030497_

Round 1

Reviewer 1 Report

The work by M.S. Silva et al is about the preparation, characterization and trypanocidal use of PLA nanoparticles of DETC. The work is interesting for the scientific community.

Please find below minor errors to be corrected before publication:

-Celsius degree symbol ( ºC) should be used correctly along the text.

-In page 3 row 98 says "Contretas et al" and should say "Contreras et al"

-Equations 1 and 2 in pages 4 and 5 should be numbered.

-In page 6 row 183 says "Masmmon" and should say "Mosmman"

-In Table 3 says "Preciptation" but should say "Precipitation"

Author Response

#Comment 1: Celsius degree symbol (ºC) should be used correctly along the text.

R: Thanks for all the comments. This issue has been revised in the new version of the manuscript.

#Comment 2: In page 3 row 98 says "Contretas et al" and should say "Contreras et al"

R: This issue has been revised in the new version of the manuscript.

#Comment 3: Equations 1 and 2 in pages 4 and 5 should be numbered.

R: This issue has been revised in the new version of the manuscript.

#Comment 4: In page 6 row 183 says "Masmmon" and should say "Mosmman"

R: This issue has been revised in the new version of the manuscript.

#Comment 5: In Table 3 says "Preciptation" but should say "Precipitation"

R: This issue has been revised in the new version of the manuscript.

Reviewer 2 Report

The manuscript "In vitro validation of antiparasitic activity of PLA-nanoparticles of sodium diethyldithiocarbamate against Trypanosoma cruzi" describes the encapsulation of sodium diethyldithiocarbamate (DETC) by poly-lactic acid (PLA) in nanoparticles and their physical characterization by DLS, SEM and AFM. PLA-DETC is a system of biodegradable nanoparticles capable of reducing toxicity caused by free DETC against cells and maintaining the antiparasitic activity.

The research has been carried out properly, the manuscript is clearly written and well balanced, the results are of interest.

Some minor points should be addressed before acceptance:

Lines 53, 67, 463: please remove the full stop before the bibliographic references.

Line 64: probably “promising” it is better than promise.

Line 77: please remove italic from “moreover”.

Lines 254, 257, 260, 308: please, use capital letter on text for figure panel.

Figure 2: please, insert letters A and B in figure as described in legend.

Line 299: please insert space after numbers.

Line 518: please insert space before reference.

Author Response

#Comment 1: Lines 53, 67, 463: please remove the full stop before the bibliographic references.

R: Thanks for all the comments. This issue has been revised in the new version of the manuscript.

#Comment 2: Line 64: probably “promising” it is better than promise.

R: This issue has been revised in the new version of the manuscript.

#Comment 3: Line 77: please remove italic from “moreover”.

R: This issue has been revised in the new version of the manuscript.

#Comment 4: Lines 254, 257, 260, 308: please, use capital letter on text for figure panel.

R: This issue has been revised in the new version of the manuscript.

#Comment 5: Figure 2: please, insert letters A and B in figure as described in legend.

R: This issue has been revised in the new version of the manuscript.

#Comment 6: Line 299: please insert space after numbers.

R: This issue has been revised in the new version of the manuscript.

#Comment 7: Line 518: please insert space before reference

R: This issue has been revised in the new version of the manuscript.

Reviewer 3 Report

This article describes the preparation and characterization of sodium diethyldithiocarbamate supported on PLA-nanoparticles. Toxicity was tested against three cells lines. The anti-parasitic activity of these materials against different strains of Trypanosoma cruzi was studied.

The English used needs to be improved. There are many grammatical mistakes. The article should be revised preferably by a native English speaker. In certain parts these language errors could lead to ambiguity or confusion for example line 412: Furthermore, the parasites when exposure to nanoparticles without DETC demonstrate the capacity increase ROS production either.

Cellular toxicity of DETC nanoparticles. Could the authors compare cell viability of free-DECT? In the new graphic the concentration of NPD would correspond to that of DECT present. In this way the same quatity of DECT (the active agent) could be directly compared,

The anti-parasitic activity results are encouraging. Why was free-DECT not tested / not presented?

Author Response

#Comment 1: The English used needs to be improved. There are many grammatical mistakes. The article should be revised preferably by a native English speaker. In certain parts these language errors could lead to ambiguity or confusion for example line 412: Furthermore, the parasites when exposure to nanoparticles without DETC demonstrate the capacity increase ROS production either.

R: Thanks for all the comments. This issue has been revised in the new version of the manuscript.

#Comment 2: Cellular toxicity of DETC nanoparticles. Could the authors compare cell viability of free-DECT? In the new graphic the concentration of NPD would correspond to that of DECT present. In this way the same quatity of DECT (the active agent) could be directly compared.

R: In the manuscript discussion session, these comparative data are discussed, as our research group has already published the cytotoxicity data of free DETC previously: J.W. de Freitas Oliveira, T.M. Torres, C.J.G. Moreno, B. Amorim-Carmo, I.Z. Damasceno, A.K.M.C. Soares, J. da Silva Barbosa, H.A.O. Rocha, M.S. Silva, Insights of antiparasitic activity of sodium diethyldithiocarbamate against different strains of Trypanosoma cruzi, Sci. Rep. 11 (2021) 1–13.

#Comment 3:The anti-parasitic activity results are encouraging. Why was free-DECT not tested / not presented?

R: In the manuscript discussion session, these comparative data are discussed, as our research group has already published the cytotoxicity data of free DETC previously: J.W. de Freitas Oliveira, T.M. Torres, C.J.G. Moreno, B. Amorim-Carmo, I.Z. Damasceno, A.K.M.C. Soares, J. da Silva Barbosa, H.A.O. Rocha, M.S. Silva, Insights of antiparasitic activity of sodium diethyldithiocarbamate against different strains of Trypanosoma cruzi, Sci. Rep. 11 (2021) 1–13.